# A Hypothesis of Gut–Liver Mediated Heterosis: Multi-Omics Insights into Hybrid Taimen Immunometabolism (*Hucho taimen* ♀ × *Brachymystax lenok* ♂)

**DOI:** 10.3390/ani16010074

**Published:** 2025-12-26

**Authors:** Mingliang Wei, Shuqi Wang, Feng Lin, Shicheng Han, Tingting Zhang, Youyi Kuang, Guangxiang Tong

**Affiliations:** 1Heilongjiang Fisheries Research Institute, Chinese Academy of Fishery Sciences, Harbin 150070, China; weimingliang@hrfri.ac.cn (M.W.); wsq15620821103@163.com (S.W.); hanshicheng@hrfri.ac.cn (S.H.); zhangtingting@hrfri.ac.cn (T.Z.); kuangyouyi@hrfri.ac.cn (Y.K.); 2College of Fisheries and Life Science, Dalian Ocean University, Dalian 116023, China; 3Sichuan Zumuzu River Basin Hydropower Development Co., Ltd., Chengdu 610059, China; linfeng233332025@163.com

**Keywords:** hybrid taimen, *Hucho taimen*, gut microbiota, transcriptome

## Abstract

Hybrid taimen (*Hucho taimen* ♀ × *Brachymystax lenok* ♂) survive gut infections better than their purebred parents, but the reason was unclear. We compared the gut bacteria and host gene expression of hybrid and pure fish reared under identical conditions. Hybrids carried more bacteria predicted to make antimicrobial and short-chain fatty acid metabolites, and their intestines and livers showed more vigorous immune and energy-supply gene activity. While these correlations suggest a beneficial gut–liver axis, future controlled experiments are needed to prove that specific microbes or genes actually cause the improved survival.

## 1. Introduction

Cold-water fish species, such as *Hucho taimen* and *Brachymystax lenok*, are keystone organisms in East Asian aquatic ecosystems [1,2,3,4]. However, their wild populations have experienced a systematic decline, leading to their classification as Vulnerable (VU) by the International Union for Conservation of Nature (IUCN) [1,4]. This decline is primarily attributed to habitat fragmentation and overfishing, which have severely impacted their reproductive success. As ecologically significant species, their preservation is critical not only for biodiversity but also as an indicator of the ecological health of major river basins, such as the Heilongjiang River [5].

In response to these conservation challenges and growing market demand, artificial breeding techniques for these species have been refined, improving their economic viability in aquaculture. Notably, the F1 generation of hybrid taimen (*H. taimen* ♀ × *B. lenok* ♂) exhibits significant heterosis, displaying growth rates 25–70% higher than purebred parents, a >50% reduction in malformation rates, and a 40% improvement in feeding efficiency for artificial diets [2,5,6]. This hybrid vigor extends beyond growth; preliminary observations from our team under identical recirculating aquaculture system (RAS) conditions suggest that hybrid taimen may possess superior disease resistance compared to H. taimen, as evidenced by a lower incidence of common diseases such as enteritis and saprolegniosis. The gut microbiota is increasingly recognized as a crucial factor influencing host health, immunity, and disease resistance in both terrestrial and aquatic animals [7]. In fish, a balanced gut microbial community plays a vital role in nutrient metabolism, pathogen exclusion, and the development of the intestinal mucosal immune system [8].

Despite these insights, the specific microbial and host molecular factors that underpin the observed heterosis in disease resistance within hybrid taimen remain poorly understood. This knowledge gap persists because previous studies have primarily focused on growth performance, leaving the underlying immunological mechanisms largely unexplored. To address this gap, researchers must rely on tools such as 16S rRNA gene sequencing to profile microbial community structure [7,8]. While it is crucial to acknowledge that functional predictions derived from such data using tools like PICRUSt2 are inherently inferential and require cautious interpretation and experimental validation, they remain indispensable for generating testable hypotheses and guiding future mechanistic studies.

Therefore, this study was designed to address this gap by combining multi-omics approaches. We hypothesize that the enhanced disease resistance observed in hybrid taimen is connected to a unique gut microbial community and specific changes in host intestinal and liver gene expression profiles. Specifically, we aim to: (1) characterize and compare the gut microbiota of hybrid taimen and purebred H. taimen; (2) DEGs in intestinal and liver tissues between the two groups; and (3) explore potential links between particular microbial taxa and host gene expression pathways related to immunity and metabolism. By achieving these goals, we aim to establish a foundational molecular profile that can inform future functional studies and selective breeding efforts to enhance disease resistance in cold-water aquaculture.

## 2. Materials and Methods

### 2.1. Experimental Methods

This study was conducted at the Wendegen Hydropower Station Fish Propagation Facility in Inner Mongolia Autonomous Region, China. To ensure accuracy and address potential tank effects, fish were raised under uniform conditions in a RAS consisting of twenty fiberglass tanks (internal dimensions: 4.0 m × 1.0 m × 0.5 m; water depth: 0.20 m). Nine tanks (designated as the ZJ group) each contained 1000 one-month-old hybrid taimen (average length = 2.20 ± 0.31 cm; average weight = 0.24 ± 0.04 g), while another nine tanks (designated as the ZL group) each held 11,000 one-month-old *H. taimen* fry (average length = 2.13 ± 0.03 cm; average weight = 0.23 ± 0.07 g). The remaining two tanks served as unstocked controls. After five months of rearing- during which incidents of enteritis and post-disease survival rates were recorded- healthy fish from each group were selected for further acclimation. Specifically, 1000 healthy fish from the hybrid taimen (average body length = 9.20 ± 0.86 cm; average body weight = 10.94 ± 2.84 g) and *H. taimen* (average body length = 9.40 ± 0.69 cm; average body weight = 10.58 ± 1.86 g) groups were transferred to separate, clean, and empty recirculating aquaculture system tanks for a 20-day acclimation period. This step helped reduce the potential effects of recent disease challenges and ensured the fish’s physiological stability before sampling. Six healthy *H. taimen* and six healthy hybrid taimen were randomly selected from three different tanks per group (two fish per tank) to minimize tank- specific bias. All data are reported as mean ± standard deviation (SD). All fish were anesthetized with MS-222. Intestinal contents from all twelve fish were collected and immediately stored at −80 °C for microbiome analysis. Concurrently, liver and intestinal tissues from three fish per group (one from each tank) were sampled and stored at −80 °C for transcriptome analysis. All samples were sent to Personal Biotechnology Co., Ltd. (Shanghai, China) for 16S rRNA sequencing and transcriptome analysis. In subsequent experiments, H. taimen and hybrid taimen are referred to as the ZL and ZJ groups, respectively. The sample size for transcriptomic analysis (*n* = 3 per group) was based on preliminary data and aligned with similar fish multi-omics studies that have successfully identified significant DEGs using DESeq2 with proper FDR control, acknowledging this as a limitation for detecting minor expression changes.

### 2.2. Gut Microbiome Analysis

Total genomic DNA was extracted from 200 mg of intestinal content per fish using the MagBeads FastDNA Kit for Soil (MP Biomedicals, Irvine, CA, USA; Cat. No. 116564384) following the manufacturer’s instructions. DNA concentration and purity were assessed using a NanoDrop™ One spectrophotometer (Thermo Fisher Scientific, Wilmington, DE, USA) and agarose gel electrophoresis (1%, *w*/*v*); only samples with A260/A280 ratios between 1.8 and 2.0 were retained for downstream analysis. The bacterial full-length 16S rRNA gene (V1–V9, ~1500 bp) was amplified using the universal primers 27F (5′-AGAGTTTGATCMTGGCTCAG-3′) and 1492R (5′-ACCTTGTTACGACTT-3′), with the forward primer carrying an 8 bp sample-specific barcode to enable multiplexing. PCR reactions (50 μL) contained 25 μL Q5^®^ High-Fidelity 2× Master Mix (NEB, Ipswich, MA, USA), 1 μL of each primer (10 μM), 2 μL template DNA (ca. 50 ng), and nuclease-free water to volume. Thermal cycling conditions were: 95 °C for 3 min; 30 cycles of 95 °C for 30 s, 55 °C for 30 s, 72 °C for 90 s; and a final extension at 72 °C for 10 min. Triplicate PCR products per sample were pooled, purified using AMPure XP beads (Beckman Coulter, Boston, MA, USA) at a bead-to-DNA ratio of 0.8×, and quantified with a Qubit dsDNA HS Assay Kit (Thermo Fisher Scientific, Wilmington, DE, USA). Purified amplicons were pooled in equimolar amounts (final concentration 2 nM) and subjected to library preparation using the SMRTbell Prep Kit 3.0 (Pacific Biosciences, Menlo Park, CA, USA) according to the manufacturer’s protocol. Full-length 16S rRNA libraries were sequenced on the PacBio Sequel II platform (Pacific Biosciences, Menlo Park, CA, USA) using Circular Consensus Sequencing (CCS) mode (CCS reads ≥ 3 passes, accuracy ≥ Q20). Raw CCS reads were demultiplexed and primers/barcode sequences were removed using cutadapt v3.5. Quality filtering, denoising, and chimera removal were performed with the DADA2 plugin in QIIME2 v2023.2, generating high-resolution amplicon sequence variants (ASVs). ASVs were taxonomically classified against the NCBI RefSeq 16S rRNA database (release 220) using the q2-feature-classifier plugin with a confidence threshold of 0.8. Alpha-diversity indices (Chao1, Shannon, Simpson, and Faith’s PD) were calculated using q2-diversity. Beta-diversity was assessed by weighted and unweighted UniFrac distances, and principal coordinates analysis (PCoA) was performed to visualize community dissimilarities. Statistical significance between groups was tested using PERMANOVA (999 permutations) with the adonis function (vegan v2.6−4). Differential abundance of taxa was determined using ANCOM-II with a cutoff of 0.05. Differential abundance of individual taxa between groups was subsequently assessed by independent-sample Student’s *t*-test (normally distributed and equal variance) or Wilcoxon rank-sum test (non-normal or unequal variance) using relative abundance matrices exported from QIIME2, with significance set at *p* < 0.05. Functional potential was predicted with PICRUSt2 v2.5.0 against the KEGG Orthology database. Pathway-level differences between groups were identified using the Kruskal–Wallis test with Benjamini–Hochberg FDR correction (adjusted *p* < 0.05).

### 2.3. Intestinal and Liver Transcriptome Analysis

Total RNA was extracted from liver and intestinal tissues using a standard TRIzol protocol. RNA integrity was assessed using an Agilent Bioanalyzer, and samples with RIN values greater than 7.0 were considered of high quality. Poly(A) + mRNA was enriched from total RNA using oligo (dT) magnetic beads and fragmented by ion shearing into approximately 300 bp segments. Using the resulting RNA as template, first-strand cDNA was synthesised with random hexamer primers and reverse transcriptase, followed by second-strand cDNA synthesis. The cDNA library was then amplified by PCR and size-selected to a target insert size of 450 bp. Library quality was evaluated using an Agilent Bioanalyzer, and both total and effective concentrations were quantified. Libraries with distinct index sequences were pooled proportionally according to effective concentration and required sequencing depth. The pooled library was diluted to 2 nM, denatured into single strands, and subjected to paired-end sequencing on the Illumina platform.

FASTQ-formatted raw sequencing data were processed with fastp (v0.22.0) to trim 3′ adapter sequences and filter out reads with average quality scores below Q20, yielding high-quality clean data. A reference-based alignment strategy was employed for transcriptome analysis. The high-quality reference genome and corresponding gene annotation file for *Hucho hucho* were retrieved from the NCBI database. A genome index was constructed using HISAT2 (version 2.1.0), and paired-end clean reads were aligned to the reference genome using the same tool. Gene-level read counts were quantified with HTSeq (v0.9.1) as raw expression values. Differential gene expression analysis was conducted between comparison groups (ZLC vs. ZJC for the intestine, ZLG vs. ZJG for the liver) using DESeq2 (version 1.38.3) in R. The DESeq2 model was designed to account for tissue type and fish species. Genes with an adjusted *p*-value (Benjamini-Hochberg FDR) < 0.05 and |log_2_FoldChange| > 1 were identified as DEGs. GSEA was performed using clusterProfiler (v4.6.0), which ranks all genes according to their differential expression levels between groups and evaluates whether predefined gene sets show significant enrichment at either extreme of the ranking to reveal activated or suppressed biological pathways. Additionally, functional enrichment analysis was performed using clusterProfiler (v4.6.0) via the hypergeometric test to identify significantly overrepresented Gene Ontology (GO) terms and KEGG pathways (adjusted *p* < 0.05) among all differentially expressed genes (DEGs), as well as upregulated and downregulated DEGs.

### 2.4. qRT-PCR

To thoroughly verify the reliability of the transcriptome sequencing data, twenty differentially expressed genes (DEGs) were selected for expression trend validation using quantitative real-time PCR (qRT-PCR). The selection was balanced: ten genes were chosen from the intestinal comparison (ZLC vs. ZJC), with five up-regulated and five down-regulated, and another ten genes from the hepatic comparison (ZLG vs. ZJG), also evenly split between up- and down-regulated. Primer sequences for these twenty candidate genes were designed with Primer-BLAST (NCBI) to span exon-exon junctions where possible and are listed in Appendix A. Primer specificity was confirmed through conventional PCR followed by Sanger sequencing of the amplicons. qRT-PCR was conducted on a CFX96 Touch Real-Time PCR Detection System (Bio-Rad) using SYBR Green Master Mix. Each reaction was performed in triplicate, and each plate included a no-template control (NTC) to monitor contamination. The amplification efficiency for each primer pair was determined by generating a standard curve from a five-point 10-fold serial dilution of cDNA; all primers showed efficiencies between 90% and 110%. The stability of the internal reference gene, β-actin, was assessed across all samples using the geNorm algorithm, confirming its suitability for normalization. The relative expression levels of target genes were calculated using the 2^−ΔΔCt^ method. The expression trends (i.e., upward or downward regulation) observed from the qRT-PCR analysis were compared with those from the RNA-seq data to evaluate the reliability of the transcriptomic results.

## 3. Results

### 3.1. Survival Rates of Intestinal Inflammation in Hybrid Taimen Versus Hucho taimen 

During aquaculture in a recirculating system, this study observed that both species of *Hucho* developed enteritis, with no other diseases present, on three separate occasions. Throughout the farming process, we recorded the survival rates of two fish species following episodes of intestinal inflammation under the same rearing conditions. It was found that the survival rate of hybrid taimen after each episode of intestinal inflammation was significantly higher than that of *H. taimen* (*p* < 0.05, Figure 1).

### 3.2. Abundance and Diversity of Microbial Communities

A total of 849,533 clean reads were obtained from the intestinal microbiota of 12 fish samples, averaging 70,796 reads per sample. After chimaera removal, 845,819 high-quality tags were clustered into 3272 amplicon sequence variants (ASVs). Rarefaction curves indicated that the sequencing depth was sufficient to capture microbial diversity across all samples (Appendix A). A Venn diagram revealed 55 core ASVs shared between groups, while *H. taimen* (ZL) and hybrid taimen (ZJ) harboured 1674 and 1543 unique ASVs, respectively (Appendix A). PCoA based on both unweighted and weighted UniFrac distances demonstrated clear separation between the intestinal microbiota of the two groups (Appendix A). No significant differences (*p* > 0.05) in α-diversity were observed between *H. taimen* and hybrid taimen under identical rearing conditions (Figure 2). In the β-diversity analysis, the PCoA (Figure 3A) and UMAP (Figure 3B) results indicated that the ZL and ZJ groups did not differ significantly in their community structures. Hierarchical clustering (Figure 3C) further revealed that the similarity among samples was primarily driven by individual variation rather than by the grouping label, thereby reinforcing the notion that there are no systematic differences between the two groups. The intra-group distance distributions for ZL and ZJ overlapped almost entirely, with comparable means and medians. Moreover, the within-group variability did not differ significantly between the two groups (*p* = 0.52 > 0.05; Figure 3D). Under the experimental conditions of this study, PERMANOVA analysis of the gut microbiota of hybrid taimen and *H. taimen* from the Tai-men population indicated that hybridization did not lead to a systematic restructuring of the microbial community composition in these salmonids (Table 1).

### 3.3. Microbial Community Composition

The intestinal microbiota of hybrid taimen (ZJ) and *H. taimen* (ZL) exhibited similar structural profiles and dominant bacterial phyla, with no significant intergroup differences observed (*p* > 0.05). The predominant phyla across all samples included *Bacillota*, *Pseudomonadota*, *Cyanobacteria*, and *Actinobacteria*, with *Bacillota* representing the most abundant phylum in both groups (Figure 4A,C and Appendix A). At the genus level, however, the two groups exhibited compositional differences. *Marinobacterium* was the most abundant genus in both ZL and ZJ, though the difference was not statistically significant (*p* > 0.05, Figure 4B,D and Appendix A). The species Sankey diagram for the ZL and ZJ groups (Figure 4G) indicates that the gut microbiota composition of the two groups is essentially similar.

### 3.4. Microbial Community Structure Shifts

To examine differences in bacterial abundance between fish species, linear discriminant analysis effect size (LEfSe) was used to compare *H. taimen* (ZL) and hybrid taimen (ZJ). The cladogram revealed clear enrichment patterns in the ZJ group (Figure 5A). At the phylum level, there were no statistically significant differences in microbial abundance between the two groups (Figure 5C,D). Using a linear discriminant analysis (LDA) score threshold of 4, the ZL group showed enrichment of seven bacterial taxa, including *Arthrobacter* and *Toxopsis* (Figure 5B,D). In contrast, the ZJ group showed a significant enrichment of Hapalosiphon and Tepidimicrobium (Figure 5B,D).

### 3.5. Functional Prediction of Gut Microbiota

Principal component analysis (PCA) revealed that the samples used to predict gut microbiota functions in the two fish species largely overlapped (Figure 6A). At the genus level, the contribution of individual taxa to the predicted functional profile closely mirrored their relative abundances (Figure 6B). Based on PICRUSt2 predictions against the KEGG database, functional profiling of the gut microbiota in both groups showed that the top three enriched pathways were primarily associated with biosynthesis of ansamycins (ko01501), valine, leucine, and isoleucine biosynthesis (ko00290), and fatty-acid biosynthesis (ko00061) (Figure 6C). Examination of the predicted KEGG pathways indicated that the top 11 taxa contributing to gut microbiota function all belonged to Metabolism (Figure 6D). Functional divergence analysis demonstrated that hybrid taimen exhibited a significantly stronger enrichment of Selenocompound metabolism (ko00450) compared with *H. taimen* (*p* < 0.05; Figure 6E).

### 3.6. Transcriptome Analysis of Liver and Intestinal Tissues

#### 3.6.1. Identification of DEGs

To clarify the molecular mechanisms behind growth differences between ZL (*H. taimen*) and ZJ (hybrid taimen) groups, RNA sequencing was performed on liver and intestinal tissues from six-month-old fish reared under the same conditions. After removing poly-N sequences, adapters, and low-quality reads, a total of 80.35 GB of high-quality clean data was obtained from six intestinal and six liver samples (Appendix A). The transcriptome data showed an average GC content of 47.47%, with Q20 and Q30 values exceeding 98.38% and 94.09%, respectively, confirming the high quality of the sequencing, which is suitable for further analysis. The clean reads were mapped to the *Hucho hucho* genome, with a mapping rate ranging from 77.54% to 87.63% (Appendix A).

DEGs were identified using thresholds of |log_2_FoldChange| > 1 and *p*-value < 0.05. A total of 8231 DEGs were detected across intestinal and hepatic comparisons between *H. taimen* (ZL) and hybrid taimen (ZJ) groups, with 860 DEGs shared among all comparisons (Figure 7A). Specifically, 4233 DEGs (2673 up-regulated and 1560 down-regulated) were identified in the intestinal comparison (ZLC vs. ZJC), and 3980 DEGs (1550 up-regulated and 2430 down-regulated) in the hepatic comparison (ZLG vs. ZJG) (Figure 7B). PCA and clustering analysis revealed clear separation among the four sample groups (ZLC, ZLG, ZJC, ZJG). Intestinal samples (ZLC and ZJC) and hepatic samples (ZLG and ZJG) were grouped, indicating tissue-specific transcriptomic profiles (Figure 7C,D). The three-dimensional PCA reveals that samples within the same experimental group cluster tightly (PC1 = 91.8%, PC2 = 3.2%, PC3 = 2.0%). In contrast, samples from different groups separate distinctly, confirming the reproducibility of the experimental design. Volcano plots (Figure 7E,F) (|log_2_FC| > 1 and *p* < 0.05) further identified key differentially expressed genes. In the ZLC vs. ZJC comparison, significantly altered genes include Fas-associated via death domain (*fasd*), dipeptidase 1 (*dpep1*), and alpha-2-macroglobulin (*a2m*). In the ZLC vs. ZJG comparison, the genes that have undergone marked changes are serine protease inhibitor (serpin), cytochrome P450 family one subfamily B member 1 (*cyp1b1*), and acyl-CoA synthetase family member 2 (*acsf2*). These genes are promising candidates for further functional validation. These results demonstrate consistent expression patterns within the same tissue types, with notable differences in expression between intestinal and hepatic tissues.

#### 3.6.2. Functional Enrichment Analysis Based on GSEA

To further characterise the phenotypic divergence between *H. taimen* and hybrid taimen under aquaculture conditions, Gene Set Enrichment Analysis (GSEA) was conducted on both GO terms and KEGG pathways using intestinal and hepatic transcriptomes. In intestinal tissues, the top five enriched GO terms were: acute inflammatory response (GO: 0002526, ES = 0.60), complement activation (GO: 0006956, ES = 0.71), protein activation cascade (GO: 0072376, ES = 0.74), regulation of complement activation (GO: 0030449, ES = 0.75), and regulation of protein activation cascade (GO: 2000257, ES = 0.77) (Figure 8A). All enrichment scores (ES) were positive, indicating higher expression of these gene sets in hybrid taimen intestines. In liver tissues, the most significantly enriched GO terms included: cholesterol biosynthetic process (GO: 0006695, ES = 0.52), secondary alcohol biosynthetic process (GO: 1902653, ES = 0.49), sterol biosynthetic process (GO: 0016126, ES = 0.46), sterol metabolic process (GO: 0016125, ES = 0.36), and cholesterol metabolic process (GO: 0008203, ES = 0.38) (Figure 8B). The positive ES values suggest elevated expression of these processes in hybrid taimen livers.

KEGG pathway analysis indicated that the most enriched pathways in intestinal tissues were: complement and coagulation cascades (ko04610, ES = 0.70), lysosome (ko04142, ES = 0.42), metabolism of xenobiotics by cytochrome P450 (ko00980, ES = 0.47), TGF-beta signaling pathway (ko04350, ES = 0.28), and valine, leucine, and isoleucine degradation (ko00280, ES = 0.27) (Figure 8C). All showed positive enrichment in hybrid taimen. In liver tissues, the key enriched KEGG pathways included: fat digestion and absorption (ko04975, ES = 0.51), fatty acid degradation (ko00071, ES = 0.32), metabolism of xenobiotics by cytochrome P450 (ko00980, ES = 0.56), PPAR signaling pathway (ko03320, ES = 0.41), and steroid biosynthesis (ko00100, ES = 0.71) (Figure 8D). The consistently positive ES values suggest increased activity of these metabolic pathways in hybrid taimen livers.

#### 3.6.3. qRT-PCR Analysis

qRT-PCR validation was performed by selecting five up-regulated and five down-regulated genes from each comparative transcriptomic group (ZLC vs. ZJC and ZLC vs. ZJC). The expression patterns of all chosen genes aligned with the transcriptome sequencing results, thus confirming the reliability of the transcriptomic data (Figure 9).

## 4. Discussion

### 4.1. A Multi-Level Synergistic Mechanism of Intestinal Immunity in Hybrid taimen

Gut microbiome analysis has become a key method for studying nutritional metabolism and immune defence mechanisms in fish [7]. The present study demonstrated that the gut microbial composition of hybrid taimen and *H. taimen* was highly similar when raised under the same aquaculture conditions. At the phylum level, the dominant intestinal microbial groups in both species included *Bacillota*, *Pseudomonadota*, *Cyanobacteria*, and *Actinobacteria* [8]. No significant differences in their relative abundances were found between hybrid taimen and *H. taimen*. Therefore, the slight differences in relative abundance are probably due to the broad taxonomic resolution used and the standardised rearing conditions.

Nevertheless, it is notable that *Cyanobacteria* were identified as the third most dominant phylum in both species. This is particularly concerning, as *Cyanobacteria* mainly exert adverse effects on fish gut health, especially in stressed or immunocompromised hosts [9,10]. As opportunistic pathogens, they compromise intestinal integrity through multiple mechanisms, including the production of toxins, induction of inflammation, disruption of the epithelial barrier, and competition for nutrients [11]. These processes collectively impair digestive and absorptive functions, potentially leading to systemic infections [12]. This pathological profile may also explain the vulnerability of *H. taimen* to enteritis even under standard aquaculture conditions. To explore this further, we conducted a more detailed taxonomic analysis of *Cyanobacteria*, which revealed significant differences in genus-level composition between hybrids and *H. taimen*. According to LefSe analysis, the relative abundance of *Hapalosiphon* was significantly higher in hybrid taimen. This *Cyanobacterial* genus is recognized for producing structurally unique alkaloids, including hapalindole-type compounds that exhibit anti-algal and antifungal activities [13]. These findings imply that although *Cyanobacteria* are present in the gut microbiota of hybrid taimen, some strains may also have antipathogenic properties.

To further explore the mechanisms behind these observations, subsequent intestinal transcriptomic analysis revealed significantly higher expression levels of elastin and collagen genes in the protein digestion and absorption pathway of hybrid taimen compared to *H. taimen*. In this context, as essential components of the extracellular matrix protein (ECM), the upregulation of these structural proteins activates the TGF-β signaling pathway [14]. Specifically, bone morphogenetic proteins (*bmps*) form complexes with BMP receptors I/II (*bmprI/II*), resulting in the phosphorylation of small mothers against decapentaplegic 1/5/8 (*smad1/5/8*) [15]. This signaling cascade promotes the differentiation of smooth muscle cells and supports intestinal tissue repair following injury. Based on these findings, we suggest that hybrid taimen have enhanced physical defence mechanisms against harmful gut bacteria compared to *H. taimen*. In addition to these structural defenses, transcriptomic analysis revealed that in hybrid taimen, genes encoding complement receptor 3 (*cr3*) and complement receptor 4 (*cr4*) within the complement and coagulation cascades are upregulated. This upregulation fosters synergistic defence between phagocytes and T cells, especially against intracellular pathogens [16,17]. At the same time, tissue damage from excessive complement activation is reduced by the downregulation of the complement 5a receptor 1 (*c5ar1*) gene expression [18].

Furthermore, the significant increase in gene expression of carboxypeptidase B2 (*cpb2*), alpha-2-macroglobulin (*a2m*), and *serpin* suppresses fibrin degradation products, thereby preventing their overstimulation of immune cells such as macrophages [19,20,21]. This reduction consequently lowers the production of pro-inflammatory cytokines like tumour necrosis factor-alpha (*tnf-α*) and interleukin-6 (*il-6*), indicating that the hybrid taimen gut has more robust innate immune regulation and a quicker response capacity [22,23]. In addition to these cytokine-mediated mechanisms, we also observed significantly increased lysosomal pathway activity in the hybrid gut. As vital components of the piscine immune system, lysosomes mediate pathogen clearance through coordinated phagocytosis and enzymatic degradation, ensuring efficient removal of invading microorganisms [24]. The hybrid taimen gut exhibits a sophisticated lysosomal network characterised by diverse lysosomal acid hydrolases—notably proteases (e.g., cathepsins), Glycosidases (e.g., beta-galactosidase), sulfatases (e.g., N-acetylglucosamine-6-sulfatase), lipases (e.g., lysophospholipase 3), and sphingomyelinases (e.g., sphingomyelin phosphodiesterase 1)- that perform coordinated substrate processing [25]. This includes the degradation of macromolecules, removal of sulfate groups, turnover of membrane phospholipids, and metabolism of sphingolipids. Through these functions, the system maintains intracellular homeostasis, prevents pathogen proliferation, and strengthens mucosal immunity, forming an integrated defensive barrier [26].

In summary, the intestinal immune advantage of hybrid taimen results from a multi-level synergistic mechanism. First, increased expression of elastin and collagen promotes barrier repair via the TGF-β pathway. Second, precise regulation of the complement system—through upregulation of *cr3* and *cr4* gene expression and downregulation of *c5ar1* gene expression—enhances pathogen clearance while also reducing excessive inflammation.

### 4.2. Potential Roles of Gut Microbiota in Shaping Immune Function of Hybrid taimen

Acknowledging the potential involvement of gut microbiota in these processes, we utilized a PICRUSt2-based KEGG functional prediction analysis to gain a preliminary understanding of their putative functional contributions in hybrid taimen. Our predictive analysis highlighted the biosynthesis of the ansamycins pathway (ko01051) as the primary functional module in the gut microbiota of both fish species. Metagenomic predictions of the gut microbiota in *Salmo salar* reared in a recirculating aquaculture system (RAS) and during the seawater transfer phase revealed a pronounced enrichment of the ansamycin biosynthesis pathway, which was even more prominent in the seawater (SW) stage than in the freshwater (FW) stage. The research noted that ansamycins are natural antimicrobial compounds capable of protecting fish against a variety of pathogens and hypothesized that this pathway is activated during environmental transition to confer resistance to bacterial infections [27]. We hypothesize that both fish species activate this pathway as a means to suppress pathogenic bacteria within their gut microbiota. It is plausible that such a function contributes to maintaining gut microbiota homeostasis by inhibiting competing pathogens via antimicrobial compounds [28]. A study has shown that commonly encountered environmental antibiotics suppress the protein activities of CYP1A and CYP3A in *Oncorhynchus mykiss* liver microsomes [29].

Interestingly, we observed the downregulation of genes involved in the metabolism of xenobiotics by the cytochrome P450 pathway in hybrid taimen, specifically cytochrome P450 family 1 subfamily A (*cyp1a*) and cytochrome P450 family 1 subfamily B member 1 (*cyp1b1*). Under identical rearing conditions and in the absence of external confounding factors, the hybrid taimen exhibited this phenomenon. It is tempting to speculate that this host response may be linked to antimicrobial compounds produced by the gut microbiota, which could interfere with the cellular uptake of exogenous toxins during Phase I metabolism. The biosynthesis of ansamycins may represent a functional role of the microbiota in generating ansamycin antibiotics (e.g., rifamycin). However, the detailed mechanisms underlying xenobiotic metabolism via the cytochrome P450 pathway in fish have not been fully elucidated; consequently, this study can only infer potential mechanisms by integrating knowledge from mammals and other taxa with our intestinal transcriptomic data. In this study, the observed down-regulation of *cyp* gene expression likely serves as a protective strategy by preventing the formation of harmful electrophilic intermediates [30]. These intermediates, if produced, would covalently bind to and damage cellular DNA and proteins [30,31]. Consequently, this metabolic blockade may facilitate the cellular efflux of the parent toxic compounds, thereby reducing their intracellular accumulation. Simultaneously, hybrid taimen upregulate the expression of Glutathione S-transferase pi 1 (*gstp1*) and Glutathione S-transferase theta 1 (*gstt1*) genes during Phase II of this metabolic pathway, thereby enhancing cellular detoxification capacity [32,33,34]. By catalyzing the conjugation of glutathione (GSH) to electrophilic compounds—including reactive intermediates produced in Phase I metabolism—this process effectively detoxifies these harmful molecules, thereby safeguarding the cell from oxidative stress and macromolecular damage [30,34]. This reaction is a key step in detoxification, facilitating the subsequent cellular efflux of these harmful species.

Notably, functional predictions indicated a significantly higher relative abundance of the selenocompound metabolism pathway (ko00450) in the gut microbiota of hybrid taimen compared to *H. taimen* (*p* < 0.05). This observation may be interpreted as evidence for a greater selenium metabolic potential within the hybrid’s microbiota. It is therefore plausible that this enhanced microbial capacity plays a contributory role in the host’s selenium absorption and utilisation efficiency. Several studies have demonstrated that feeding fish diets supplemented with various selenium sources results in a significant increase in plasma and tissue selenium concentrations. In contrast, selenium is either undetectable or present only in trace amounts in the feces, indicating that the fish intestine absorbs the majority of the ingested selenium [35]. Moreover, both animal experiments and small-scale clinical trials have demonstrated that moderate selenium supplementation—particularly with organic selenium compounds or selenium nanoparticles—can alleviate intestinal inflammation and improve mucosal barrier function [36].

This study hypothesizes that the higher survival rate of hybrid taimen trout under intestinal inflammation, compared with *H. taimen*, may be associated with the gut microbiota’s selenium-compound metabolism pathway. Accordingly, we propose a mechanistic model—drawn from mammalian and other non-fish systems—that links gut microbial functions, specifically the biosynthesis of the ansamycin pathway (ko01051) and the selenocompound metabolism pathway (ko00450), to host intestinal interactions in fish. When the fish intestine absorbs selenium, the uptake of microbiota-metabolized selenium compounds by intestinal epithelial cells may promote selenocysteine (Sec) production, potentially leading to the synthesis of key antioxidant selenoproteins, such as glutathione peroxidase (*gpx*) [37]. This mechanism further strengthens cellular redox capacity through GSH binding [30,34]. Moreover, our findings show that in hybrid taimen, the production of prostaglandin D2 (*pgd2*), a lipid signalling molecule catalysed by hematopoietic Prostaglandin D Synthase (*hpgds*) [38,39]. These molecules effectively reduce inflammatory responses, facilitate efficient peroxide clearance, maintain intracellular redox homeostasis, and safeguard cellular membranes and DNA against oxidative damage. However, this study did not perform quantitative analyses of ansamycins or selenium levels in the intestine or intestinal contents. Consequently, the proposed model is purely hypothetical and is based solely on the multi-omics data generated in the present investigation.

### 4.3. Omics-Guided Hypothesis: Gut–Liver Axis Ignition of Cholesterol Synthesis as a Metabolic Edge in Hybrid taimen

Within fish intestines, *Bacillota* (formerly known as *Firmicutes*) and *Pseudomonadota* (previously referred to as *Proteobacteria*) are two of the most common and coexisting bacterial phyla [40]. Following the official reclassification by the International Committee on Systematics of Prokaryotes (ICSP), the phylum names *Firmicutes* and *Proteobacteria* remain valid. However, *Pseudomonadota* has become the accepted term for the group previously referred to as Proteobacteria [41]. In this study, *Bacillota* and *Pseudomonadota* were the dominant phyla, consistent with previous reports on fish intestinal microbiota [41].

*Bacillota* formed the dominant bacterial group in both hybrid taimen and *H. taimen*. Evidence increasingly shows that *Bacillota* can degrade proteins and complex carbohydrates, producing SCFAs such as butyrate, propionate, and acetate [42]. These metabolites have various effects on host metabolism by binding to receptors on enteroendocrine cells. They also provide energy to intestinal epithelial cells, enhancing nutrient absorption [43]. The relative abundance of *Tepidimicrobium* was found to be higher in hybrid taimen than in *H. taimen*. This observation suggests that this genus may be a significant contributor to the gut’s metabolic activity. Through its fermentative metabolism of various organic compounds, including proteins, peptides, amino acids, and carbohydrates, it is plausible that *Tepidimicrobium* generates metabolites such as acetate, hydrogen, and carbon dioxide, as well as essential precursors for the synthesis of short-chain fatty acids, thereby potentially influencing the gut environment [44].

We also found that functional prediction of the gut microbiota identified the valine, leucine, and isoleucine biosynthesis pathway (ko00290) in both species. This finding suggests a conserved microbial potential for producing branched-chain amino acids (BCAAs) [45]. Although this study did not quantify the levels of branched-chain amino acids (BCAAs) or short-chain fatty acids (SCFAs) in the liver, intestine, or intestinal contents, transcriptomic analysis unexpectedly revealed that the hybrid taimen exhibited a markedly greater activation of the intestinal valine, leucine, and isoleucine degradation pathway (ko00280) relative to the hybrid taimen. The observed upregulation of 3-hydroxy-3-methylglutaryl-CoA lyase-like 1 (*hmgcll1*) suggests that hybrid taimen possess an enhanced capacity for breaking down excess BCAAs. In mammals, this process may involve a preferential catabolism of leucine [46]. This metabolic adaptation promotes the conversion of acetoacetate into acetyl-CoA via Acetoacetyl-CoA Synthetase (AACS)-mediated catalysis, facilitating entry into the tricarboxylic acid (TCA) cycle and ultimately supporting cholesterol biosynthesis and related anabolic pathways [47,48].

Simultaneously, we predicted that the gut microbiota possesses the capacity for fatty-acid biosynthesis (ko00061). At the genus level, we also detected the presence of *Clostridium* in the intestinal communities of both *Hucho* strains. *Clostridium* is recognized as the most critical anaerobic gut bacterium in mammals for long-chain fatty-acid (LCFA) degradation and short-chain fatty-acid (SCFA) production, capable of converting dietary or host-derived LCFAs into SCFAs that confer health benefits to the host [49,50,51]. When SCFAs are transported to the liver via the portal vein, the hepatic fatty acid degradation pathway is activated [52]. Upregulation of the gene expression of acyl-CoA synthetase long-chain family member 1a (*acsl1a*) and acyl-CoA synthetase long-chain family member 3a (*acsl3a*) enhances the gene expression of hydroxyacyl-CoA dehydrogenase trifunctional multienzyme complex subunit alpha B (*hadhab*). Acetyl-CoA acyltransferase 1 (*acaa1*), improving their capacity to desulfurise SCFAs and ultimately catalyse their conversion into acetyl-CoA [53,54,55,56]. This acetyl-CoA pool further stimulates the steroid biosynthesis pathway, resulting in increased cholesterol synthesis [57]. In this study, the expression of genes such as farnesyl diphosphate farnesyltransferase 1 (*fdft1*), squalene synthase (*sqlea*), lateral splicing site (*lss*), cytochrome P450 51A1 (*cyp51*), and transmembrane 7 superfamily domain-containing protein 2 (*tm7sf2*) was significantly upregulated in the liver of hybrid taimen individuals. These genes play critical roles in cholesterol synthesis and the metabolism of steroid precursors. Their upregulation may enhance the activity of key enzymes involved in the steroid biosynthesis pathway, thereby promoting the synthesis of steroid precursors and activating the entire biosynthetic pathway [58,59,60]. These results indicate that, under identical rearing conditions, hybrid taimen exhibits a more substantial hepatic cholesterol biosynthetic capacity than *H. taimen*. As the obligatory precursor for piscine steroidogenesis, cholesterol is essential for preserving membrane structural integrity and dynamic fluidity, properties that directly enable proper immune cell activation, proliferation, and migratory capacity [60]. At the same time, hybrid taimen increase the gene expression of peroxisome proliferator-activated receptor γ (pparγ) and retinoic acid receptor (RAR) within the PPAR signaling pathway, thereby further enhancing the transcription of *scd-1* and *cyp27*. This activation promotes lipid metabolism and energy generation, thus supplying essential metabolic substrates for immune responses [61,62].

Based on the association between the predicted functional profiles of the gut microbiota and the host transcriptome, we derived and proposed a hypothetical integrated liver-gut axis immune network model (Figure 10) for hybrid taimen, aiming to elucidate the potential synergistic mechanisms of the gut-liver axis in this hybrid species.

## 5. Conclusions

This study describes hybrid taimen (*Hucho taimen* ♀ × *Brachymystax lenok* ♂) as a naturally occurring “proof-of-concept” model where subtle, taxonomically minor shifts in the gut microbiome align with broad, coordinated changes in intestinal and liver gene networks that control immunity, antioxidant capacity, and energy metabolism. Instead of depending on a single protective taxon or a single immune pathway, the advantage of hybrid taimen seems to come from a multi-layered gut–liver axis: (i) increase in bacteria believed to produce antimicrobial alkaloids (*Hapalosiphon*) and short-chain fatty acid precursors (*Tepidimicrobium*); (ii) increased expression of complement/lysosome/TGF-β genes that strengthen mucosal barrier function; and (iii) up-regulated hepatic PPAR and cholesterol biosynthesis genes that help maintain membrane fluidity and steroid-based immunity. We suggest that these microbial and host modules create a positive feedback loop—SCFAs and selenium metabolites reach the liver via the portal vein, boost PPAR signaling, and support the energy needs of an active immune system without causing excessive inflammation. Because our data are correlative, proving causality will require reciprocal microbiota transplantation, CRISPR knock-out of host pathways (e.g., *smad1/5/8*, *pparγ*), and metabolomic tracing of SCFAs or selenium compounds under controlled bacterial challenges. Long-term studies across developmental stages, different temperature conditions, and various pathogens will clarify the stability of the hybrid-associated microbiome when environmental factors fluctuate. From an industry standpoint, the research offers two practical, immediate options. First, the hybrid-associated taxa and predicted functions (ko00450, ko01501) can be developed into probiotic mixes or bioactive feed additives for purebred taimen or other cold-water salmonids, potentially simulating the hybrid phenotype without diluting the genetics of wild stocks. Second, the identified intestinal and hepatic gene signatures represent the first molecular markers for selectively breeding disease-resistant broodstock, thereby accelerating genomic selection programs that currently focus primarily on growth traits. Ultimately, understanding the key points of the hybrid gut–liver axis should allow for precision aquaculture strategies that improve animal welfare, reduce antibiotic use, and help conserve endangered taimen populations.

## Figures and Tables

**Figure 1 animals-16-00074-f001:**
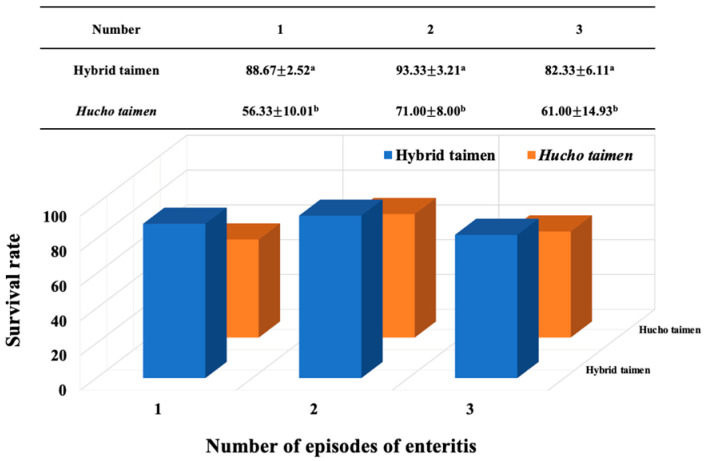
Survival rate comparison between hybrid taimen and *H. taimen* following three episodes of enteritis. Note: Different lowercase letters denote statistically significant differences between groups.

**Figure 2 animals-16-00074-f002:**
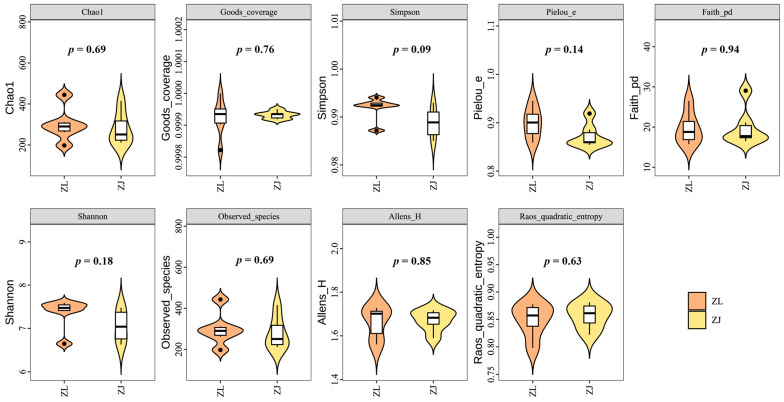
α-Diversity of intestinal microbiota in *H. taimen* and hybrid taimen under identical rearing conditions.

**Figure 3 animals-16-00074-f003:**
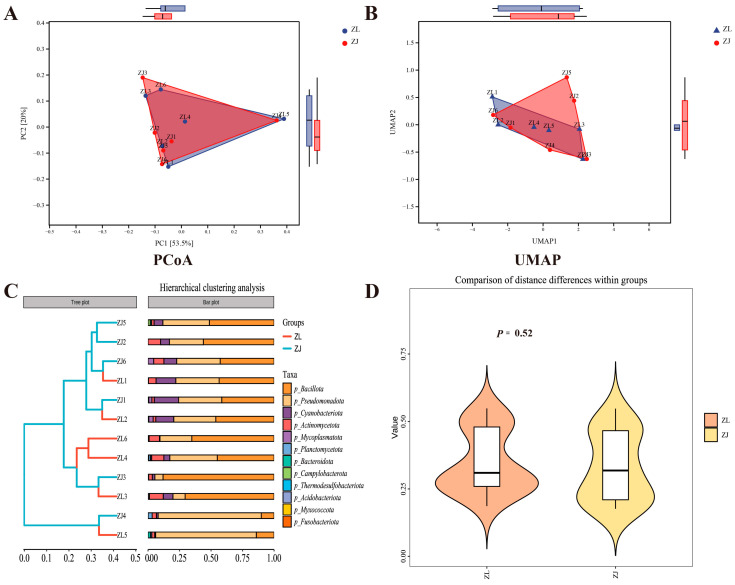
β-diversity analysis (**A**) Principal Coordinates Analysis (PCoA); (**B**) Unweighted Pair Group Method with Arithmetic Mean (UPGMA) analysis; (**C**) Hierarchical clustering analysis; (**D**) Within-group variation analysis.

**Figure 4 animals-16-00074-f004:**
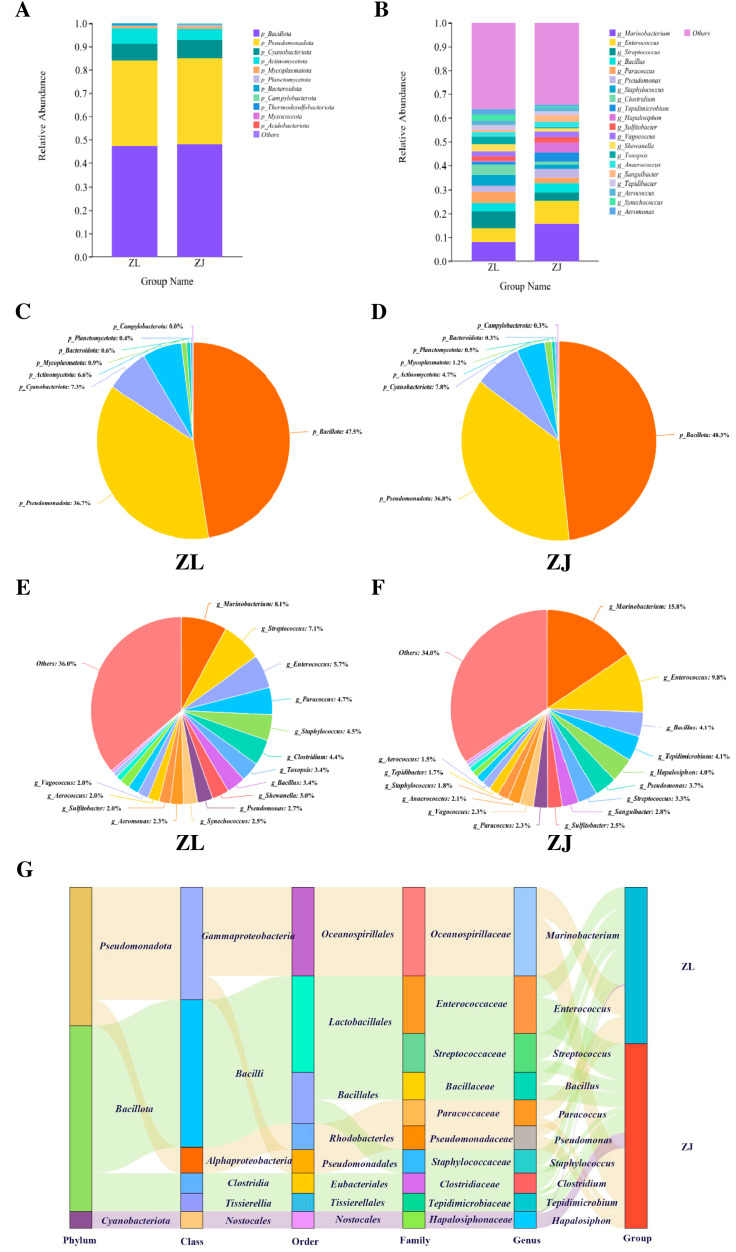
Relative abundance of dominant bacterial phyla (**A**) and genera (**B**) in the intestinal microbiota of *H. taimen* (ZL) and hybrid taimen (ZJ); Major phylum composition in ZL (**C**) and ZJ (**D**); Major genus composition in ZL (**E**) and ZJ (**F**); Scatter plot illustrating microbial community structure between ZL and ZJ (**G**).

**Figure 5 animals-16-00074-f005:**
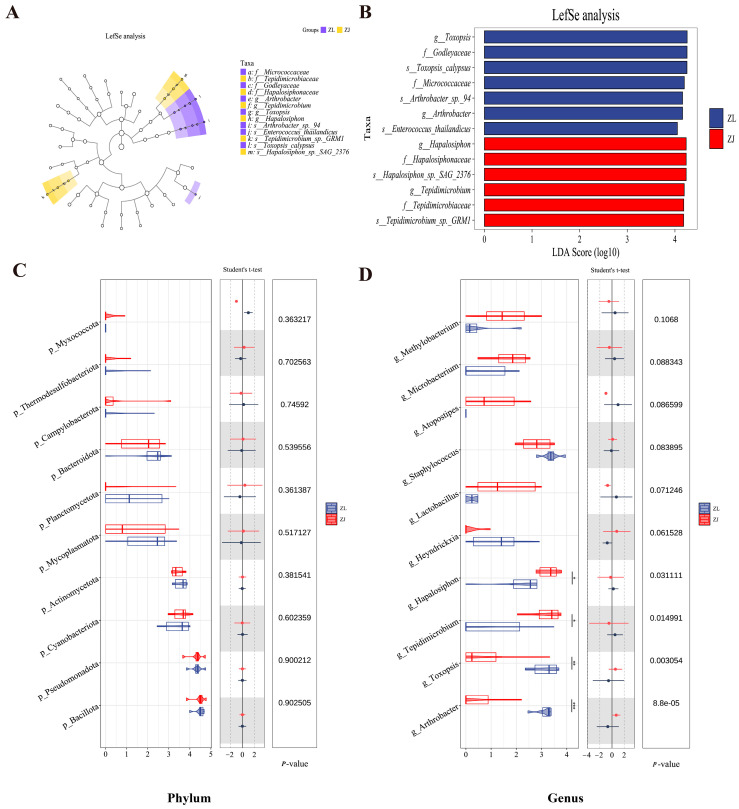
Identification of differential ASVs in gut bacterial communities between *H. taimen* and hybrid taimen. (**A**) LEfSe cladogram displaying the phylogenetic distribution of discriminative features. (**B**) Histogram of LDA scores for bacterial taxa that are significantly enriched (LDA score > 4.0). (**C**) Comparative abundance of gut microbiota in *H. taimen* (ZL) and hybrid Taimen (ZJ) at the phylum level. (**D**) Comparative abundance of gut microbiota in *H. taimen* (ZL) and hybrid taimen (ZJ) at the genus level. Note: In Figures C and D, * indicates significant, ** indicates highly significant, and *** indicates extremely significant.

**Figure 6 animals-16-00074-f006:**
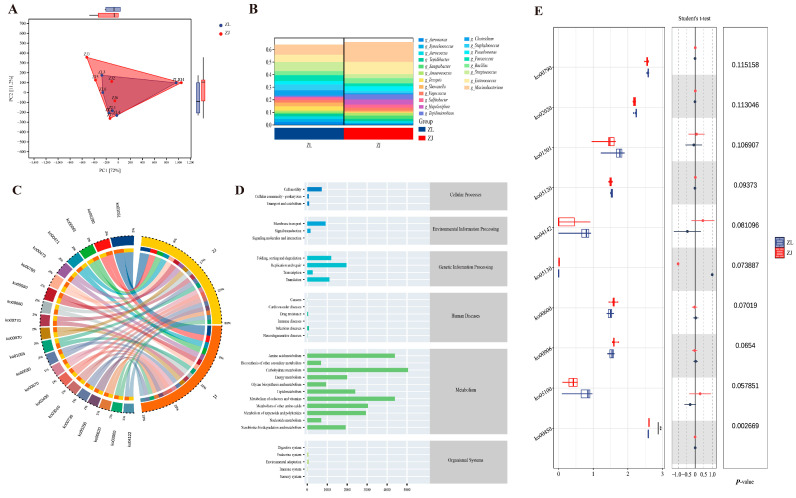
Predicted functional profiling of the gut microbiota in hybrid taimen and *H. taimen*. (**A**) Principal Component Analysis (PCA); (**B**) Species contribution analysis; (**C**) PICRUSt2-KEGG Circos plot; (**D**) KEGG pathway bar chart; (**E**) Differential abundance analysis of KEGG-predicted microbial functions. Note: In (**E**), ** indicates statistical significance.

**Figure 7 animals-16-00074-f007:**
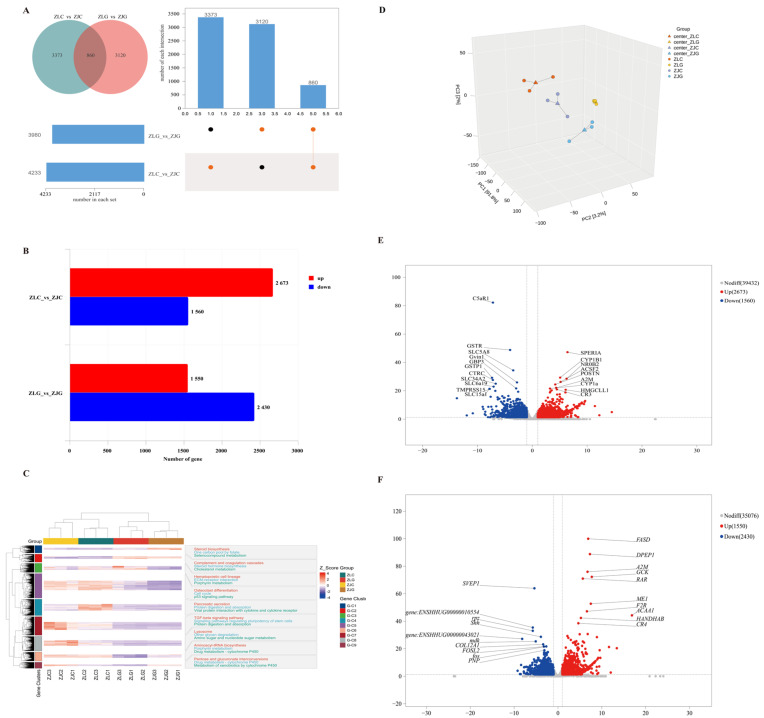
Transcriptomic analysis of liver and intestinal tissues in *H. taimen* (ZL) and hybrid taimen (ZJ). (**A**) Venn diagram and UpSet plot illustrating the distribution of differentially expressed genes (DEGs) across comparison groups. (**B**) Bar chart summarizing the number of DEGs in each comparison. (**C**) Hierarchical clustering heatmap displaying gene expression patterns in intestinal and hepatic tissues. (**D**) Principal component analysis (PCA) of transcriptome profiles from both tissue types. (**E**) Volcano plot of differentially expressed genes (DEGs) for ZLC vs. ZJC. (**F**) Volcano plot of differentially expressed genes (DEGs) for ZLG vs. ZJG. Note: In the figure A, the sets involved in each intersection are displayed beneath the bar chart. Points connected by lines denote the sets that participate in the intersection: orange points indicate sets that are part of a given intersection, while black points indicate sets that are not involved in that intersection.

**Figure 8 animals-16-00074-f008:**
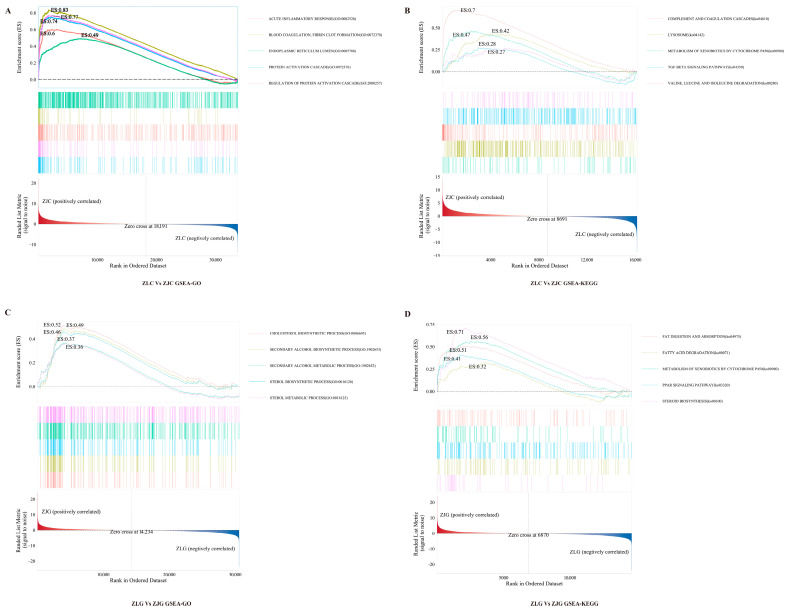
Functional enrichment analysis of differentially expressed genes between *H. taimen* (ZL) and hybrid taimen (ZJ) based on GSEA. The vertical axis shows functional terms or pathways, while the horizontal axis indicates gene ratio. Colour intensity reflects adjusted *p*-values, and dot size represents the number of DEGs. (**A**) GSEA-GO enrichment analysis of ZLC vs. ZJC; (**B**) GSEA-KEGG analysis of ZLC vs. ZJC; (**C**) GSEA-GO analysis of ZLG vs. ZJG; (**D**) GSEA-KEGG analysis of ZLG vs. ZJG.

**Figure 9 animals-16-00074-f009:**
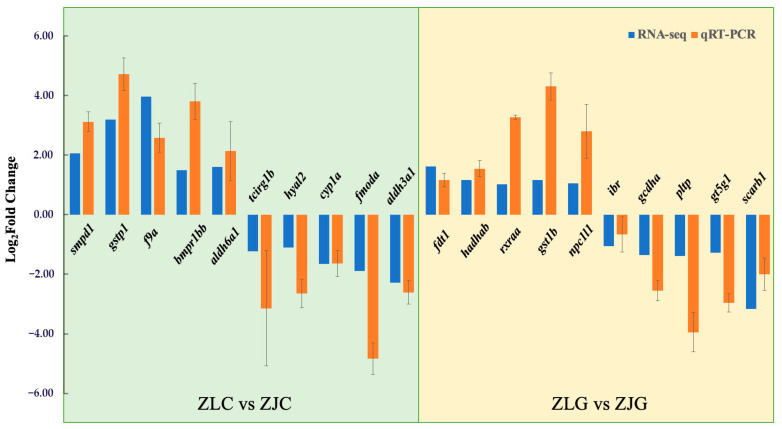
Validation of gene expression trends from transcriptome sequencing by qRT-PCR.

**Figure 10 animals-16-00074-f010:**
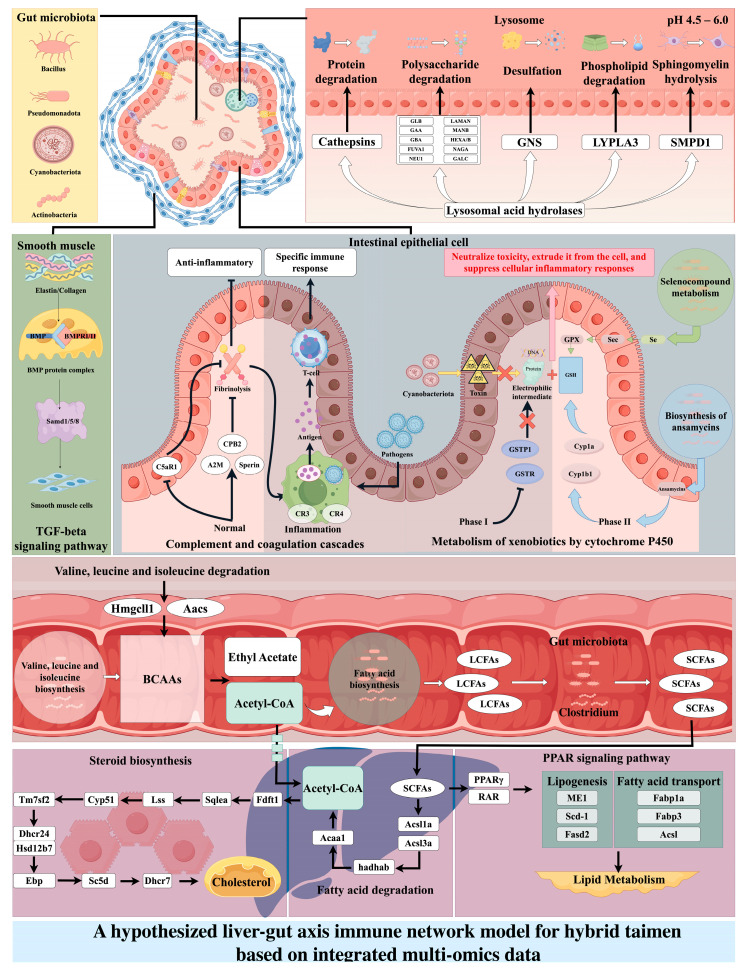
A Proposed Hypothetical Model of the Gut-Liver Immune Network in Hybrid Taimen, Integrating Multi-omics Evidence. Note: This schematic summarizes a working hypothesis derived from correlative multi-omics data. It proposes potential interactions between gut microbiota functions and host gut-liver axis communication that may underlie enhanced disease resistance in hybrid taimen. Causal relationships require experimental validation.

**Table 1 animals-16-00074-t001:** PERMANOVA.

Table	Df	SumsOfSqs	MeanSqs	F.Model	R^2^	Pr (>F)
Treat1	1	0.0255	0.0255	0.3952	0.038	0.785
Residuals	10	0.6444	0.0644	NA	0.962	NA
Total	11	0.6699	NA	NA	1	NA

Note: Treat1: *H.* taimen and hybrid taimen; Df: Degrees of Freedom; SumsOfSqs: Sum of Squares; R: R-squared: Pr (>F): Probability value associated with F-test; NA: No Data.

## Data Availability

The transcriptome datasets have been fully deposited in the National Center for Bioinformation (CNCB-NGDC) as required. The project accession number is PRJCA051472, and the data are now publicly available at https://ngdc.cncb.ac.cn (accessed on 11/12/2025). The gut microbiota datasets are stored in the Genome Sequence Archive (GSA). Due to research confidentiality and sample privacy considerations, these data are not publicly accessible at this time. If reviewers or readers require access to these data, please contact the corresponding author directly, and we will provide the files promptly.

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
