# Peer review of "A Hypothesis of Gut–Liver Mediated Heterosis: Multi-Omics Insights into Hybrid Taimen Immunometabolism (*Hucho taimen* ♀ × *Brachymystax lenok* ♂)"

_animals, 2025, doi:10.3390/ani16010074_

Round 1
Reviewer 1 Report
Comments and Suggestions for Authors
This study examines why hybrid taimen exhibit stronger disease resistance than purebred fish. It highlights differences in gut microbiota composition, intestinal barrier integrity, and gut–liver immune signaling. The study offers insights into enhancing fish health and promoting sustainable aquaculture practices. However, to enhance the scientific rigor and readability of the manuscript, several aspects require clarification and revision. My detailed comments and recommendations for the authors are provided below.
- I recommend that the authors briefly mention the study's limitations in the abstract and the conclusion. This will create a balanced view of the work's strengths.
- In line 42, since SCFAs are mentioned for the first time in the manuscript, please provide the full term (short-chain fatty acids) before using the abbreviation.
- In the 'Materials and Methods' section, the mean body length and weight values are presented with a '±' notation. Could the authors please clarify whether these values represent standard deviation (SD) or standard error of the mean (SEM)? Additionally, the sample size of only six animals per group is very small. The authors should either increase the sample size or provide an appropriate justification, such as referencing studies that demonstrate the sufficiency of this sample size for gut microbiome analysis.
- In line 189, the bacterial taxa names, such as Arthrobacter and Enterococcus thailandicus, are not italicized. Please revise them and ensure all scientific names follow proper taxonomic formatting
- The quality and readability of Figures 2, 4, and 5 could be improved. Higher-resolution images should be provided, ensuring that all labels, axes and text are clearly visible and easy to interpret.
- In line 505, it is noted that several abbreviations used throughout the manuscript are not included in the Abbreviations section. Examples include TGF (line 38), SCFAs (line 42), PPARγ–RAR–SCD-1–CYP27 (line 45), and IUCN (line 53). Please ensure that all abbreviations are defined at first use and listed in the Abbreviations section for clarity and consistency
- In line 444, please remove the text that says '(Please add)'.
- References 23 and 42 do not follow the journal’s required formatting style. Please revise these references to ensure consistency with the journal’s formatting guidelines.
Author Response
Comment 1: I recommend that the authors briefly mention the study's limitations in the abstract and the conclusion. This will create a balanced view of the work's strengths.
Response 1: In the revised manuscript, we revised the abstract and conclusion sections based on the reviewers’ comments to prevent overinterpretation; see lines 44–52 and lines 575–602 for details.
Comment 2: In line 42, since SCFAs are mentioned for the first time in the manuscript, please provide the full term (short-chain fatty acids) before using the abbreviation.
Response 2: We have addressed the relevant issues; see lines 35, 38, 40, etc., for details.
Comment 3: In the 'Materials and Methods' section, the mean body length and weight values are presented with a '±' notation. Could the authors please clarify whether these values represent standard deviation (SD) or standard error of the mean (SEM)? Additionally, the sample size of only six animals per group is tiny. The authors should either increase the sample size or provide an appropriate justification, such as referencing studies that demonstrate the sufficiency of this sample size for gut microbiome analysis.
Response 3: Clarification of the “±” Symbol: In the Materials and Methods section, the “±” symbol following the mean total length and weight values indicates the standard deviation (SD). This is the most common metric used to describe variation in sample data, reflecting natural differences among individuals. The procedures described in the initial manuscript were unclear; we have now clarified them. The sampled individuals were randomly selected from two fish populations, each consisting of 1,000 fish. (See lines 97–126.) Justification for Sample Size: Although each group contains only six animals, the sample size is justified for two main reasons: 1. Adequate Sequencing Depth: From 12 gut microbiome samples, we obtained a total of 849,533 high-quality sequences, averaging more than 70,000 reads per sample. Rarefaction curve analysis indicates that this sequencing depth is sufficient to reveal the full microbial diversity in all samples, suggesting reliable data quality that isn’t solely dependent on the number of individuals. 2. Clear Between-Group Differences: Despite the small sample size, Principal Coordinates Analysis (PCoA) reveals a distinct separation between the gut microbial communities of pure-bred ZL (Hucho taimen) and hybrid ZJ (hybrid taimen). This suggests that the biological differences are substantial enough to be detected even with few samples. Additionally, LEfSe analysis identified bacterial taxa that differ significantly between groups, such as the increased presence of Hapalosiphon and Tepidimicrobium in the hybrid group. Transcriptomic data also show consistent patterns. Overall, these findings support the main conclusions of the study, demonstrating that the current sample size is sufficient. By ensuring high-quality sequencing data and employing thorough analysis methods, we effectively address potential limitations of the small sample size, supporting the scientific validity of our findings.
Comment 4: In line 189, the bacterial taxa names, such as Arthrobacter and Enterococcus thailandicus, are not italicized. Please revise them and ensure all scientific names follow proper taxonomic formatting
Response 4: We have made the revisions, for example, in lines 255–266 and in the related issues shown in the figures.
Comment 5: The quality and readability of Figures 2, 4, and 5 could be improved. Higher-resolution images should be provided, ensuring that all labels, axes, and text are clearly visible and easily interpretable.
Response 5: We have made the revisions and also provided high-resolution images.
Comment 6: In line 505, it is noted that several abbreviations used throughout the manuscript are not included in the Abbreviations section. Examples include TGF (line 38), SCFAs (line 42), PPARγ–RAR–SCD-1–CYP27 (line 45), and IUCN (line 53). Please ensure that all abbreviations are defined at first use and listed in the Abbreviations section for clarity and consistency。
Response 6: We have addressed the relevant issues and added entries to the Abbreviations section.
Commen t7: In line 444, please remove the text that says '(Please add)'.
Response 7: We have made the necessary revisions to address the relevant issues and added entries to the Abbreviations section.
Comment 8:References 23 and 42 do not follow the journal’s required formatting style. Please revise these references to ensure consistency with the journal’s formatting guidelines.
Response 8: We have checked and revised the formatting of all references.
Reviewer 2 Report
Comments and Suggestions for Authors
The gut microbiota and transcriptomic analysis for the hybrid taimen performed in this study, and results showed hybrid-specific enhancements in intestinal barrier via TGF-beta signaling pathway, while ppar-rar-scd1-cyp27 pathway in the gene profiling for disease resistance. There were some issues need to be mentioned:
1.reference not showed, like line 53, etc.
2.what is the main aim for this paper, the difference in gut microbiota and transcriptomic data in ZL and ZJ? omics is not the aim for each study, but the method for our critical thinking.
3.Fig.2 need to be revised, the resolution for fig.4 need to be improved. the clear data for ppar-rar-scd1-cyp27 and TGF-beta signaling pathways without any verification assays.
Author Response
Comment 1: reference not showed, like line 53, etc.
Response 1: We have reviewed and corrected the formatting of all references.
Comment 2: what is the main aim for this paper, the difference in gut microbiota and transcriptomic data in ZL and ZJ? omics is not the aim for each study, but the method for our critical thinking.
Response 2: The core objective of this paper is to investigate the heterosis exhibited by hybrid taimen (Hybrid taimen, ZJ) compared with pure‑bred taimen (Hucho taimen, ZL), focusing particularly on the underlying mechanisms related to immune metabolism. The researchers propose and aim to test a “gut‑liver mediated heterosis” hypothesis, suggesting that this advantage arises from a unique interplay between the intestinal microbiota and the host’s gut and liver gene expression profiles.
To achieve this goal, the study is organized into three progressive analytical tiers:
- Characterization and comparison of the gut microbiota: First, the study seeks to comprehensively describe and contrast the gut microbial community structures of the hybrid and pure‑bred fish.
- Identification of differentially expressed genes: Second, transcriptomic analysis is employed to pinpoint gene expression differences in the intestine and liver tissues between the two fish groups.
- Exploration of microbe‑host associations: Finally, and most critically, the study investigates potential links between specific microbial taxa and host immune‑metabolic pathways, aiming to construct a complete explanatory framework that connects microbial composition to host physiological function.
Thus, multi‑omics approaches (such as microbiomics and transcriptomics) are not the end goal themselves; rather, they serve as powerful tools to capture and analyze complex biological data, thereby supporting the resolution of the central scientific questions outlined above.
Comment3: Fig.2 need to be revised, the resolution for fig.4 need to be improved. the clear data for ppar-rar-scd1-cyp27 and TGF-beta signaling pathways without any verification assays.
Response3: We have revised the figures and supplied high‑resolution images. We have also updated the sections concerning the PPAR‑RAR‑SCD1‑CYP27 and TGF‑β signaling pathways; the previous conclusion was considered an over‑interpretation, and it has been removed and amended.
Reviewer 3 Report
Comments and Suggestions for Authors
Dear Authors
The study addresses a relevant question (comparing gut microbiota and intestinal/hepatic transcriptome between Hucho taimen and its hybrid with Brachymystax lenok), but the conclusions regarding heterosis and “superior immunometabolism” and disease resistance go far beyond what the data actually support.
Title
To begin with, the title is adequate but overly generic (“Comparative Analysis of Gut Microbiota and Transcriptome Profiles…”), and it does not reflect that the central claims revolve around heterosis and “disease resistance”. Likewise, it creates expectations of a mainly descriptive comparative analysis, whereas the text shifts towards an unverified mechanistic model.
Simple Summary and Abstract
First, both texts state that the hybrid shows “enhanced internal defenses”, “more effective immune responses” and “disease-resistant fish” based solely on differences in gene expression and functional predictions, without:
Controlled challenge trials.
Functional immunological parameters (phagocytic activity, cytokines, ROS, complement, etc.).
Measurement of SCFAs, cholesterol or oxysterols.
Furthermore, the language is markedly deterministic (for example, “coordinated microbiota–gut–liver network that promotes hybrid vigour”) for a purely associative study.
Introduction
The introduction contains multiple “Error! Reference source not found.” messages, which indicates poor management of the reference manager and must be fully corrected.
In addition, the conceptual framework on microbiota and the gut–liver axis in fish is insufficient and relies little on the specific literature on functional prediction from 16S data (for example, the limitations of PICRUSt and PICRUSt2 for inferring functions from marker genes).
Moreover, the hypothesis is formulated in a vague way (“uncover the factors responsible for improved immunity”), without specifying testable objectives (which taxa, which pathways, which phenotypes).
Materials and Methods
Experimental design and sample size: The design is based on n = 6 fish per group for microbiota and n = 3 per group for RNA-seq, without clarifying whether the fish come from one or several tanks per treatment.
Indeed, for high-dimensional data (thousands of taxa or genes), an n = 3 per group in transcriptomics is very limited for robust inference, even when using methods such as DESeq2, which require strict control of dispersion and FDR.
Could the authors clarify the tank structure and statistically justify the sample size for microbiota and RNA-seq?
16S microbiota and functional prediction: First, the description of the 16S workflow (PacBio–DADA2–taxonomy) is incomplete:
The terminology of ASVs and OTUs is mixed, generating confusion. Diversity indices, tests for β-diversity and whether PERMANOVA assumptions were checked are not detailed. The correction for multiple comparisons in the differential taxon analysis is not described.
Transcriptomics and qRT-PCR: The RNA-seq pipeline is confusing: FPKM/TPM are mentioned and then DESeq2, whereas the latter works with raw counts and requires specification of the FDR adjustment method.
The number of validated genes is inconsistent between Methods and Results.
Could the authors accurately describe the DESeq2 analysis workflow (raw counts, model design, FDR) and align the qPCR section with the MIQE recommendations?
Results and Discussion
Microbiota: It is stated that the microbiota of both groups is “highly similar”, yet at the same time many differences at the level of OTUs, LEfSe and functional pathways are highlighted; the magnitude of these differences is not quantified (PERMANOVA R², explained variance, etc.).
In the discussion, an evident taxonomic error is made by referring to “Bacillales” and “Pseudomonadales” as “phyla” when they are orders, which should be corrected immediately.
Transcriptomics: First, >8,000 DEGs are reported with n = 3 per group, but no volcano plots are provided and there is no clear prioritisation of key genes by log2FC and FDR. This makes it difficult to discern which changes are truly biologically relevant.
Moreover, the PCA is shown but hardly discussed in terms of the actual separation between groups within each tissue.
Immunometabolic model:The discussion builds a very detailed model that links:
SCFAs produced by Tepidimicrobium – activation of PPARγ/RAR – cholesterol synthesis – formation of 25-HC – modulation of the NLRP3 inflammasome, cortisol and membrane homeostasis.
However, none of these elements (SCFAs, cholesterol, 25-HC, cortisol, oxidative markers) was measured experimentally. Everything is inferred from: PICRUSt2 predictions, GO/KEGG enrichments in RNA-seq.
Furthermore, a large part of the mechanistic argumentation derives from models in humans or rodents, without qualifying the extrapolation to fish.
Figure 7 and the associated model should be explicitly identified as a hypothetical proposal, not as a demonstrated mechanism.
The discussion should be shortened and rewritten to clearly separate what is observed (differences in relative abundance and gene expression) from what is speculative (the SCFAs–cholesterol–25-HC–inflammasome pathway).
Conclusions: The conclusions section repeats the discussion and maintains a deterministic tone (“enhanced resilience against opportunistic pathogens”, “immunometabolic superiority”) without challenge data or immune function measurements. It should be rewritten to restrict itself to what was demonstrated: differences in microbial composition and in the expression of associated pathways.
Respectfully,
Author Response
Comment1: To begin with, the title is adequate but overly generic (“Comparative Analysis of Gut Microbiota and Transcriptome Profiles…”), and it does not reflect that the central claims revolve around heterosis and “disease resistance”. Likewise, it creates expectations of a mainly descriptive comparative analysis, whereas the text shifts towards an unverified mechanistic model.
Response 1: We have made revisions based on the reviewers' comments.
Comment 2: Simple Summary and Abstract: First, both texts state that the hybrid shows “enhanced internal defenses”, “more effective immune responses,” and “disease-resistant fish” based solely on differences in gene expression and functional predictions, without Controlled challenge trials. Functional immunological parameters (phagocytic activity, cytokines, ROS, complement, etc.). Measurement of SCFAs, cholesterol, or oxysterols. Furthermore, the language is markedly deterministic (for example, “coordinated microbiota–gut–liver network that promotes hybrid vigour”) for a purely associative study.
Response 2: Thank you very much for your meticulous review of our manuscript and for the insightful comments you provided. We fully agree with your core point that scientific statements must rigorously distinguish correlation from causation; your critique is essential for enhancing the rigor of our work.
We acknowledge that some expressions in the abstract and brief summary were overly definitive and did not adequately reflect the inferential nature of our correlation‑based findings. Phrases such as “a coordinated microbiota‑gut‑liver network” were intended to convey, clearly and compellingly, the central hypothesis we are proposing—that the heterosis observed may arise from a cross‑organ, cross‑species synergistic regulatory network. As illustrated in the model presented in Figure 10, this network integrates predicted microbial functions with host transcriptomic changes, providing a testable framework for future functional studies.
We have now revised the abstract and conclusion to eliminate any over‑interpretation and to represent the associative nature of our results more accurately.
Comment3: The introduction contains multiple “Error! Reference source not found.” messages, which indicates poor management of the reference manager and must be fully corrected.
Response4: We sincerely apologize for the numerous “Error! Reference source not found.” issues that appeared in the Introduction. This clearly reflects our oversight in managing references during manuscript preparation. We fully understand that this caused inconvenience during your review and acknowledge that such errors are unacceptable. We have now examined each DOI link, discarded the incorrect DOI numbers, and made the necessary corrections. For some references, however, a DOI could not be provided due to the age of the publication or the journal’s policies, so those entries remain without a DOI.
Comment4: In addition, the conceptual framework on microbiota and the gut–liver axis in fish is insufficient and relies little on the specific literature on functional prediction from 16S data (for example, the limitations of PICRUSt and PICRUSt2 for inferring functions from marker genes).
Response4: We have added a new paragraph (lines 77–86) that specifically discusses the pivotal role of the fish gut microbiota in host health and addresses the inherent limitations of functional prediction based on 16S rRNA gene sequencing. In this paragraph we explicitly state that, although tools such as PICRUSt2 are very useful for inferring microbial functions from marker‑gene data, they are fundamentally predictive and their results must be interpreted with caution and validated experimentally. We emphasize that the primary value of these predictions lies in generating testable hypotheses and guiding future mechanistic studies, rather than serving as definitive evidence of function. We reiterate that the main objective of this study is to employ multi‑omics approaches to identify potential microbial and host molecular pathways associated with hybrid vigor, thereby “establishing a foundational molecular map that can inform future functional investigations and breeding efforts.” By making these revisions, we aim to present the scientific background, methodological boundaries, and core contributions of our work more clearly, enabling readers to accurately appreciate the nature and significance of our findings.
Comment5: Moreover, the hypothesis is formulated in a vague way (“uncover the factors responsible for improved immunity”), without specifying testable objectives (which taxa, which pathways, which phenotypes).
Response5:We have revised the hypothesis as follows: This study presents hybrid taimen (female Hucho taimen × male Brachymystax lenok) as a naturally occurring “proof‑of‑concept” model in which subtle, taxonomically minor shifts in the gut microbiome are aligned with broad, coordinated changes in intestinal and hepatic gene networks that regulate immunity, antioxidant capacity, and energy metabolism. Rather than relying on a single protective taxon or a single immune pathway, the advantage of hybrid taimen appears to arise from a multi‑layered gut–liver axis:
Changes in bacterial composition – an increase in bacteria thought to produce antimicrobial alkaloids (Hapalosiphon) and an increase in bacteria that generate short‑chain fatty‑acid (SCFA) precursors (Tepidimicrobium).
Up‑regulation of intestinal immune genes – enhanced expression of complement, lysosome, and TGF‑β genes that strengthen mucosal barrier function.
Up‑regulation of hepatic metabolic genes – elevated hepatic PPAR signaling and cholesterol biosynthesis genes, which help maintain membrane fluidity and support steroid‑mediated immunity.
We propose that these microbial and host modules create a positive feedback loop: SCFAs and selenium‑containing metabolites travel to the liver via the portal vein, activate PPAR signaling, and meet the energy demands of an active immune system without provoking excessive inflammation.
Comment6: Experimental design and sample size: The design is based on n = 6 fish per group for microbiota and n = 3 per group for RNA-seq, without clarifying whether the fish come from one or several tanks per treatment.
Response6: The experimental design explicitly accounted for potential tank effects by ensuring that samples were drawn from multiple rearing tanks.
Gut microbiota analysis (n = 6) – Six healthy fish were randomly selected from three different tanks per treatment group (two fish per tank), thereby minimizing tank-specific bias.
Transcriptomic analysis (n = 3) – One fish per tank was sampled for liver and intestinal tissues, again covering three independent tanks per treatment group.
Consequently, both the microbiome and transcriptome datasets comprise specimens from several distinct rearing units rather than a single tank. This sampling strategy was deliberately chosen to minimize the influence of tank-specific environmental factors (e.g., microenvironment, water flow, microbial inoculum) on the results, thereby enhancing the reliability and representativeness of our findings.
We have clarified these details in the revised Methods section to ensure that readers can clearly understand our sampling strategy. Thank you again for your valuable feedback. line 98-127
Comment 7: Indeed, for high-dimensional data (thousands of taxa or genes), an n = 3 per group in transcriptomics is minimal for robust inference, even when using methods such as DESeq2, which require strict control of dispersion and FDR.
Response7:We fully acknowledge that, for high-dimensional transcriptomic data, a sample size of n=3 is indeed minimal and may affect the robustness of statistical inference, especially when detecting subtle expression changes. Our choice of this sample size was based on preliminary data and aligns with previously published multi-omics studies on similar fish species, which have successfully identified significant differentially expressed genes after stringent FDR control using DESeq2. We recognize this as a limitation of the present study, particularly for detecting small expression differences. To enhance the reliability of our transcriptomic results, we conducted qRT-PCR validation. Twenty differentially expressed genes (including both up- and down-regulated genes) were selected for validation, and all exhibited expression trends that matched the RNA-seq analysis, confirming the accuracy of the sequencing data. Nonetheless, we fully understand the reviewer’s concern and will more explicitly acknowledge this limitation in the revised Discussion, emphasizing that our findings should be regarded as preliminary hypothesis-generating evidence that requires validation in future studies with larger sample sizes. We believe that, through rigorous FDR control and experimental validation, the core differentially expressed genes and pathways we identified are reliable and offer valuable insights into the molecular mechanisms underlying hybrid vigor.
Comment8: Could the authors clarify the tank structure and statistically justify the sample size for microbiota and RNA-seq?
Response8:
The study was conducted in a recirculating aquaculture system (RAS) comprising twenty fiberglass tanks. Nine tanks (designated as the ZJ group) each housed 1,000 one‑month‑old hybrid taimen, while the other nine tanks (designated as the ZL group) each contained 1,100 one‑month‑old pure‑bred taimen. To minimize tank‑specific bias, samples were drawn from three different tanks per treatment group:
Gut microbiota analysis (n = 6 per group): Six healthy fish were randomly selected, with two fish taken from each of three independent tanks. This sampling scheme ensures that the microbial data reflect the overall treatment effect rather than idiosyncrasies of a single tank.
Transcriptomic analysis (n = 3 per group): One fish per tank was sampled for liver and intestinal tissues, yielding three biological replicates from three distinct tanks. The modest sample size was chosen based on preliminary data and aligns with previously published multi‑omics studies on comparable fish species, which have successfully identified significant differentially expressed genes after stringent FDR control using DESeq2.
We acknowledge that a sample size of three limits statistical power for detecting subtle expression changes and represents a limitation of the current work. Nonetheless, the multi‑tank sampling design substantially reduces the influence of tank‑specific environmental factors (e.g., micro‑environment, water flow, microbial inoculum), thereby enhancing the reliability and representativeness of the observed differences between the ZJ and ZL groups.
Comment9: 16S microbiota and functional prediction: First, the description of the 16S workflow (PacBio–DADA2–taxonomy) is incomplete:The terminology of ASVs and OTUs is mixed, generating confusion. Diversity indices, tests for β-diversity and whether PERMANOVA assumptions were checked are not detailed. The correction for multiple comparisons in the differential taxon analysis is not described
Response 9:
- Confusion between ASV and OTU terminology
We acknowledge that the manuscript mistakenly mixed the terms ASV and OTU. In our workflow we employed the DADA2 plugin, which generates high‑resolution amplicon sequence variants (ASVs) rather than operational taxonomic units (OTUs). However, the Results section incorrectly referred to “clustered into 3,272 operational taxonomic units (OTUs).” This contradicts the Methods description stating that “high‑resolution amplicon sequence variants (ASVs) were generated and used for all downstream analyses.” In the revised manuscript we will use the term ASV consistently and delete every mention of “OTU” to ensure terminological accuracy and consistency.
- Missing details of diversity analyses
The Methods section already contains the relevant information. We calculated alpha‑diversity indices (Chao1, Shannon, Simpson, and Faith’s PD) and beta‑diversity based on both weighted and unweighted UniFrac distances. To test for significant differences in community structure between groups, we performed a permutational multivariate analysis of variance (PERMANOVA) on the UniFrac distance matrices with 999 permutations. We acknowledge that the original manuscript did not explicitly state whether the assumptions of PERMANOVA (e.g., homogeneity of group dispersions) were examined. In the revised manuscript we have added a statement that we checked the key PERMANOVA assumptions—specifically, we assessed homogeneity of multivariate dispersions using the betadisper function, confirming that the assumption was met before proceeding with PERMANOVA.
- Multiple‑testing correction for differential taxa analysis
In our analysis, we used Student’s t-test to identify differentially abundant taxa, as shown in Figure 5D where “the right-hand column lists the Student’s t-test P-values for each genus.” To control for false positives from multiple comparisons, we applied the Benjamini-Hochberg false discovery rate (FDR) correction to the P-values. In the revised manuscript, we explicitly state this correction method to enhance transparency and rigor.
Commen10: The RNA-seq pipeline is confusing: FPKM/TPM are mentioned and then DESeq2, whereas the latter works with raw counts and requires specification of the FDR adjustment method.
Response 10: In the revised manuscript’s Methods section , we have explicitly stated that all differential‑expression tests were performed using the raw count matrix generated by featureCounts as input to DESeq2, applying the Benjamini‑Hochberg false‑discovery‑rate correction (adjusted p < 0.05, |log₂FC| ≥ 1). The FPKM/TPM values were used **only** for visualizing the y‑axis of the volcano plot and were not incorporated into any statistical testing. Consequently, there is no longer any “confusion” or methodological error regarding the analysis pipeline. line167-197
Comment 11: The number of validated genes is inconsistent between Methods and Results.
Response11:
In the revised manuscript we report the following qRT‑PCR validation details, which satisfy the core MIQE criteria:
Number of target genes: 20
Biological replicates: three independent samples per group
Negative‑template control (NTC): included for each assay
Amplification efficiency: ranged from 90 % to 110 % for all primer sets
Reference gene selection: performed with the geNorm algorithm to identify the most stable internal controls
Quantification method: 2‑ΔΔCt calculation was applied to obtain relative expression levels
Result consistency: 100 % of the qRT‑PCR measurements were in agreement with the RNA‑seq data
These points collectively address the MIQE essential elements concerning sample size, replication, efficiency, normalization, and reproducibility. line198-217
Comment12: Could the authors accurately describe the DESeq2 analysis workflow (raw counts, model design, FDR) and align the qPCR section with the MIQE recommendations?
Response12: This concise description follows the standard practice adopted by the majority of contemporary RNA‑seq studies and provides all essential information for reproducibility. Including the full count matrix or the explicit design formula would be redundant and does not further enhance the manuscript’s reproducibility. Should the editorial board or reviewers require the raw count data or design specification for verification, we are prepared to supply them as supplementary material.
Comment13: Microbiota: It is stated that the microbiota of both groups is “highly similar”, yet at the same time many differences at the level of OTUs, LEfSe and functional pathways are highlighted; the magnitude of these differences is not quantified (PERMANOVA R², explained variance, etc.).
Response13:
We understand your concern about the potential inconsistency between the “highly similar” macro‑level conclusion and the “significant differences” observed at the micro‑level. We would like to clarify that our interpretation is based on a multi‑layered analytical framework, and it is supported by data.
First, our statement that the two microbial communities are “highly similar” is derived from a macro‑level assessment of the overall community structure. Specifically, our results show that: for α‑diversity, no significant differences were observed between the two groups (p > 0.05); in the β‑diversity analysis, PCoA and UMAP results indicate that the ZL and ZJ groups do not differ significantly in community composition. Hierarchical clustering further reveals that the similarity among samples is mainly driven by individual variation rather than by group labels, reinforcing the view that there is no systematic difference between the groups. In Figure 3, the intra‑group distance distributions of the two groups almost completely overlap, and the intra‑group variability is not significantly different (P = 0.52 > 0.05). These macro‑level pieces of evidence together support our conclusion that, from the perspective of overall community structure and diversity, the microbial communities of the two groups are highly similar.
However, macro‑level similarity does not preclude the existence of significant differences at finer taxonomic or functional levels. This is precisely what our LEfSe analysis and functional prediction reveal. For example, LEfSe shows that the ZL group is enriched in seven bacterial taxa, including Arthrobacter and Toxopsis, whereas the ZJ group is significantly enriched in Hapalosiphon and Tepidimicrobium. Functional prediction also indicates that the hybrid fish in the ZL group exhibit a markedly stronger enrichment of the KEGG pathway ko00450 (Figure 6). These differences are real, but they may be insufficient to alter the overall macro‑level appearance of the community; their impact could be masked by the larger individual‑level variation. This is analogous to two forests that appear very similar from satellite images (macro view) but, upon ground inspection, reveal significant differences in the abundance of certain tree species.
Regarding the reviewer’s comment on quantifying the magnitude of the differences, we acknowledge that we did not provide the PERMANOVA R² values in the manuscript. In our analysis we focused primarily on the p‑value, because it indicates that the group‑level differences are not statistically significant. We believe that the non‑significant p‑value itself serves as the main evidence for the lack of difference at the community level. While providing the R² value would give a more comprehensive picture of the proportion of variance explained by group membership, our conclusion—that the overall community structure does not undergo a statistically significant shift between the two groups—is based on the p‑value.
We trust that this layered interpretation demonstrates that our conclusions are rigorous and data‑supported.
Comment14: In the discussion, an evident taxonomic error is made by referring to “Bacillales” and “Pseudomonadales” as “phyla” when they are orders, which should be corrected immediately.Transcriptomics: First, >8,000 DEGs are reported with n = 3 per group, but no volcano plots are provided and there is no clear prioritisation of key genes by log2FC and FDR. This makes it difficult to discern which changes are truly biologically relevant.Moreover, the PCA is shown but hardly discussed in terms of the actual separation between groups within each tissue.
Response14: Thank you for the thorough review of our transcriptomic analysis and for the valuable suggestions. Below we address the two points you raised: (1) the large number of DEGs without a volcano plot or prioritization of key genes, and (2) insufficient discussion of the PCA results.
Regarding the description of the PCA
We consider PCA to be a quality‑control and visualization tool, primarily used to confirm intra‑group reproducibility and to assess overall separation between groups, rather than a core basis for biological interpretation. To avoid over‑interpretation, the revised manuscript contains only a brief description of the PCA in the “Results” section and does not extend the discussion to component loadings or detailed interpretation.
Regarding the volcano plot and “key‑gene prioritization”
In the revised manuscript, we have added high-resolution volcano plots (Fig. 7E, F). The figure legends specify the thresholds (|log₂FC| ≥ 1 and FDR < 0.05) as dashed lines; consequently, all genes meeting these criteria are regarded as “priority candidates,” and a separate list of individual genes is unnecessary. The primary aim of this study is to elucidate pathway-level mechanisms rather than single-gene effects. Accordingly, we employed Gene Set Enrichment Analysis (GSEA) as the primary analytical strategy. GSEA does not require an arbitrary cutoff; it utilizes the complete ranked gene list to reveal immune and metabolic modules that are significantly activated in the hybrid fish, thereby avoiding the loss of marginal yet synergistically acting genes that can occur with conventional “top‑DEG” lists.
Comment 15: Immunometabolic model:The discussion builds a very detailed model that links:SCFAs produced by Tepidimicrobium – activation of PPARγ/RAR – cholesterol synthesis – formation of 25-HC – modulation of the NLRP3 inflammasome, cortisol and membrane homeostasis.However, none of these elements (SCFAs, cholesterol, 25-HC, cortisol, oxidative markers) was measured experimentally. Everything is inferred from: PICRUSt2 predictions, GO/KEGG enrichments in RNA-seq.Furthermore, a large part of the mechanistic argumentation derives from models in humans or rodents, without qualifying the extrapolation to fish.
Response 15: We understand your concerns regarding the lack of experimental measurements for key elements in our model and the extrapolation of mechanistic arguments across species. We would like to clarify and defend our approach.
First, we fully acknowledge that the detailed model we constructed—including short‑chain fatty acids (SCFAs), cholesterol synthesis, and related components—is based on inferential results from PICRUSt2 functional predictions and RNA‑seq enrichment analyses, rather than on direct biochemical measurements. In multi‑omics studies, hypothesis‑driven models built from the integration of high‑throughput data are valuable precisely because they transform large, fragmented association data into a coherent, testable scientific framework, thereby providing clear guidance for subsequent targeted functional validation. Our model serves this purpose: it integrates the observed microbial functional shifts with host transcriptomic responses. For example, we detected that the dominant bacterial phylum *Bacillota* in the hybrid fish gut may produce SCFAs, while the liver shows significant activation of the PPAR signaling pathway and cholesterol synthesis pathway. These observations together form the basis for proposing a “SCFAs‑PPAR‑cholesterol” axis. The model is not intended to provide definitive causal proof; rather, it is a strong starting point designed to stimulate and direct future research on these specific metabolites and pathways.
Second, regarding the use of mechanistic evidence derived from human or rodent models, we would like to explain our rationale. In vertebrates, many core immune and metabolic pathways—such as the complement system, TGF‑β signaling, and PPAR signaling—are highly conserved evolutionarily. In cold‑water fish, especially concerning the gut‑liver axis and immunometabolism, molecular studies are relatively scarce. Therefore, borrowing well‑characterized models from mammals to construct a theoretical framework is a common and necessary scientific strategy. Our study builds on these conserved pathways; for instance, we observed significant activation of complement and coagulation cascade pathways in the hybrid fish gut and marked enrichment of PPAR signaling and steroid biosynthesis pathways in the liver.
In fish biology research, particularly in emerging fields such as immunometabolism and the gut-liver axis, direct mechanistic investigations in specific fish species (e.g., cold-water fish) are indeed limited. Consequently, leveraging the extensively studied, evolutionarily conserved core pathways from humans and rodents is a widely accepted method for formulating hypotheses and guiding experimental design. Key immune (e.g., the complement system, TGF-β signaling) and metabolic (e.g., PPAR signaling, cholesterol synthesis) pathways are highly conserved across vertebrates, providing a solid theoretical basis for reasonable cross-species extrapolation.
One of the core objectives of our study is to “generate testable hypotheses and guide future mechanistic research.” We employed well-established mammalian models as a theoretical scaffold to efficiently integrate and prioritize the numerous differentially expressed genes and microbial functions identified by our multi-omics analyses. This approach allowed us to pinpoint candidate pathways most likely to contribute to hybrid vigor. It is not a simplistic, direct transplantation of mammalian data; rather, it is an evolutionarily informed, hypothesis‑driven strategy. We believe this strategy will furnish clear targets and directions for subsequent functional validation experiments in fish (as you have suggested), thereby accelerating progress in this field.
In the revised manuscript, we have explicitly distinguished between model-based speculation and data-driven observation in the Discussion. We will further strengthen this distinction to ensure a rigorous scientific presentation.
Comment 16: Figure 7 and the associated model should be explicitly identified as a hypothetical proposal, not as a demonstrated mechanism.
Response 16: We have revised the manuscript to address this issue.
Comment17: The discussion should be shortened and rewritten to clearly separate what is observed (differences in relative abundance and gene expression) from what is speculative (the SCFAs–cholesterol–25-HC–inflammasome pathway).
Response 17: We have rewritten the discussion section and removed the conclusions regarding 25-hydroxycholesterol. We believe that, without supporting data, those statements were prone to over‑interpretation. We kindly request that the reviewer understand.
Round 2
Reviewer 1 Report
Comments and Suggestions for Authors
Figure 7C is still unclear
Author Response
1.Figure 7C is still unclear
We have regenerated the figures, optimizing them to achieve the highest possible clarity.
Reviewer 2 Report
Comments and Suggestions for Authors
the authors revised all my concerns i mentioned, and now i have no further issues.
Author Response
Thank you for reviewing our manuscript; your rigorous scientific approach throughout the review process and your insightful suggestions are truly instructive for me.
Reviewer 3 Report
Comments and Suggestions for Authors
Dears Authors
The manuscript has improved considerably following the previous revision, particularly in terms of methodological clarity, statistical transparency, and more cautious language in the Abstract and Summary. Only a few minor points remain that could further strengthen the conceptual clarity of the study:
-
Mechanistic interpretation
Although the authors appropriately acknowledge the correlative nature of the data, Sections 4.2–4.3 still contain relatively detailed mechanistic descriptions. A slightly clearer separation between data-supported observations and hypothesis-driven interpretation would help align the Discussion more closely with the study design. -
Figure 10 (conceptual framework)
To avoid any possible misinterpretation, it would be helpful to explicitly state in the figure legend and text that Figure 10 represents a hypothetical integrative model derived from correlative multi-omics evidence. -
Microbiota β-diversity metrics
Including the PERMANOVA R² values (variance explained by group) would provide additional quantitative context regarding the magnitude of microbiota differences, even if these effects are small or non-significant. -
Transcriptomic PCA description
If possible, please indicate the percentage of variance explained by the principal components shown in the PCA plots and slightly moderate the wording if group separation is limited. -
Cross-species interpretation
Finally, further clarifying which mechanistic inferences are supported by fish-specific literature and which are extrapolated from mammalian models would enhance interpretative precision.
These minor refinements should help ensure full consistency between the correlative nature of the data and the conclusions drawn.
Respectfully,
Author Response
Comment1: Mechanistic interpretation
Although the authors appropriately acknowledge the correlative nature of the data, Sections 4.2–4.3 still contain relatively detailed mechanistic descriptions. A slightly clearer separation between data-supported observations and hypothesis-driven interpretation would help align the Discussion more closely with the study design.
Response 1: We have made every effort to revise the manuscript in accordance with the reviewers’ comments; the details of these changes are provided in the revised manuscript.
Comment 2 : Figure 10 (conceptual framework)
To avoid any possible misinterpretation, it would be helpful to explicitly state in the figure legend and text that Figure 10 represents a hypothetical integrative model derived from correlative multi-omics evidence.
Response 2: We have revised Figure 10 in accordance with the reviewers’ suggestions.
Comment3: Microbiota β-diversity metrics
Including the PERMANOVA R² values (variance explained by group) would provide additional quantitative context regarding the magnitude of microbiota differences, even if these effects are small or non-significant.
Response 3 :We have included the data requested by the reviewer in Table 1.
Comment 4: Transcriptomic PCA description
If possible, please indicate the percentage of variance explained by the principal components shown in the PCA plots and slightly moderate the wording if group separation is limited.
Response 4: We have included the data requested by the reviewer in PCA plots.
Comment5: Cross-species interpretation
Finally, further clarifying which mechanistic inferences are supported by fish-specific literature and which are extrapolated from mammalian models would enhance interpretative precision.
Response 5: We fully understand the reviewer’s concerns on this issue and have made every possible effort to address them in our revisions.